# Analysis of secondary metabolite gene clusters and chitin biosynthesis pathways of *Monascus purpureus* with high production of pigment and citrinin based on whole-genome sequencing

**Song Zhang**[1], **Xiaofang Zeng**[2], **Qinlu Lin**[1], **Jun Liu**[1,3]*

**1** National Engineering Research Center of Rice and Byproduct Deep Processing, Central South University of Forestry and Technology, Changsha, Hunan, China, **2** College of Light Industry and Food Sciences, Zhongkai University of Agriculture and Engineering, Guangzhou, Guangdong, China, **3** Hunan provincial Key Laboratory of Food Safety Monitoring and Early Waring, Changsha, Hunan, China

\* liujundandy@csuft.edu.cn

**Data Availability Statement:** All relevant data are within the paper and its Supporting Information files.

## Abstract

*Monascus* is a filamentous fungus that is widely used for producing *Monascus* pigments in the food industry in Southeast Asia. While the development of bioinformatics has helped elucidate the molecular mechanism underlying metabolic engineering of secondary metabolite biosynthesis, the biological information on the metabolic engineering of the morphology of *Monascus* remains unclear. In this study, the whole genome of *M. purpureus* CSU-M183 strain was sequenced using combined single-molecule real-time DNA sequencing and next-generation sequencing platforms. The length of the genome assembly was 23.75 Mb in size with a GC content of 49.13%, 69 genomic contigs and encoded 7305 putative predicted genes. In addition, we identified the secondary metabolite biosynthetic gene clusters and the chitin synthesis pathway in the genome of the high pigment-producing *M. purpureus* CSU-M183 strain. Furthermore, it is shown that the expression levels of most *Monascus* pigment and citrinin clusters located genes were significantly enhanced via atmospheric room temperature plasma mutagenesis. The results provide a basis for understanding the secondary metabolite biosynthesis, and constructing the metabolic engineering of the morphology of *Monascus*.

## Introduction

Fermented products of *Monascus* spp. have been widely used in the food and pharmaceutical industry for more than 2000 years [1]. As the secondary metabolite produced by *Monascus* spp., *Monascus* pigments (MPs) are a mixture of azaphilones mainly composed of three colors (yellow, orange, and red) pigments, which possess various bioactivities, such as antimicrobial, anticancer, anti-inflammatory, and anti-obesity [2, 3]. Nowadays, due to the potential risks of allergies, carcinogenesis, and teratogenesis of synthetic pigments, natural MPs are widely used

**Funding:** This work was supported by the National Natural Science Foundation of China (No. 32101906), Natural Science Foundation of Hunan Province (No. 2021JJ31146); Open Project Program of the Hunan Provincial Key Laboratory of Food Safety Monitoring and Early Waring (No. 2021KFJJ02), Education Department of Scientific Research Project of Hunan Province (No. 20B619) and Education Department of Postgraduate Research and Innovation Project of Hunan Province (No. CX20210863. The funders had no role in study design, data collection and analysis, decision to publish, or preparation of the manuscript.

**Competing interests:** The authors have declared that no competing interests exist.

as food colorants and are well recognized by consumers [4, 5]. In addition, MPs have other applications in the pharmaceutical, textile, and cosmetics industries. Traditionally, MPs are mainly produced by solid-state fermentation (SSF) with rice as the substrate for high pigment concentration [6]. However, submerged fermentation (SF) was more widely applied in the industrial production at present due to high pigment production efficiency, an easy-to-control fermentation process, and avoidance of contamination [7, 8].

Different species of *Monascus* spp. have been isolated for the biosynthesis of various secondary metabolites. In general, *M. fuliginosus* [9, 10], *M. ruber* [11, 12] and *M. pilosus* [13–15] have a strong capacity to produce monacolin K. Nevertheless, *M. purpureus* is the most predominant microorganisms for the efficient production of MPs because of its high efficiency to produce pigments [16–18]. With the development of whole-genome sequencing (WGS) technology, the complete sequence analysis of *Monascus* has been used to reveal the chromosome evolution, regulatory mechanisms, and functional genes of *M. purpureus*, which lays the foundation for the production of secondary metabolites and biological researches. In 2015, Yang et al. published the first sequence information of *M. purpureus* YY-1, with a genome size of 24.1 Mb and a total of 7491 predicted genes. WGS analysis predicted the gene clusters related to pigment biosynthesis in *M. purpureus* YY-1 and explained the smaller size of the *M. purpureus* genome than that of related filamentous fungi, indicating a significant loss of genes [19]. Kumagai et al. reported the genome sequence information of the high pigment-producing *M. purpureus* GB-01 strain, with a genome size of 24.3 Mb and 121 chromosomal contigs [20]. Liu et al. identified the key genes (*ERG*4A and *ERG4*B) for ergosterol biosynthesis in *M. purpureus* LQ-6 (genome size: 26.8 Mb, 8596 protein-coding genes). Knocking out the *ERG4* gene improved the permeability of the cell membrane and secretion of intracellular pigments; it also changed the morphology of *M. purpureus* LQ-6 in SF broth [21]. Although numerous studies on the morphological changes of *Monascus* in SF have been performed, the biological information on the metabolic engineering of morphology of *Monascus* remains unknown [22–24].

Hyphae of filamentous fungi in SF mainly exist in three morphological forms, including free mycelia, mycelial pellets, and mycelial clumps [25], and the difference of metabolites is probably due to the different morphology of hyphae. The mycelium pellet is the optimal morphology for glucoamylase production by *Aspergillus niger*, while the fermentation production of citric acid is more biased to the mycelial morphology [26]. The *ve*A gene globally regulates the propagation mode, mycelial growth, environmental tolerance, and secondary metabolites of fungi [27, 28]. Muller et al. disrupted the biosynthesis of chitin and changed the morphology of *Aspergillus oryzae* by regulating the transcription level of the chitin synthase gene *chs*B, and studied the relationship between morphology and α-amylase biosynthesis [29]. RNA interference technology has been applied to silence the expression of the chitin synthase gene *chs*4 in *Penicillium chrysogenum*, reducing the mutant growth rate, aggregation of dispersed hyphae into the mycelium, and increased penicillin production [30]. With the in-depth study of different phenotype mutants, it is found that the cell wall is an ideal target for morphological control. However, the differences were exsited in the encoding genes of chitin synthase and the regulation of chitin synthase on morphology in different fungi [31]. Furthermore, the specific encoding genes of chitin synthase and biosynthesis pathway of chitin in *M. purpureus* is still unclear.

*M. purpureus* CSU-M183 is a high pigment-producing industrial preparation strain obtained by atmospheric room temperature plasma (ARTP) mutation system. In this study, the whole genome of strain CSU-M183 was sequenced using the single-molecule real-time (SMRT, PacBioRS II) DNA sequencing and Illumina next-generation sequencing (NGS) platforms. We also investigated the molecular expression effects of ARTP mutagenesis on the secondary metabolic synthesis of *Monascus* by RT-qPCR. The results showed a comprehensive

prediction of biosynthetic gene clusters (BGCs) for secondary metabolites and the biosynthetic pathway of chitin in *M. purpureus* CSU-M183. We expect this will provide a better strategy in morphological metabolic engineering of *Monascus*, for the industrial production of the secondary metabolites via submerged fermentation.

# Materials and methods

## Fungal strains, culture media, and growth conditions

*M. purpureus* CSU-M183 (CCTCC M 2018224, China Central for Type Culture Collection (CCTCC), Wuhan, China) was obtained using the ARTP mutation system from the parent strain *M. purpureus* LQ-6 (CCTCC M 2018600) [32]. Strains was cultivated on potato dextrose agar (PDA) and potato dextrose broth (PDB) medium at 30˚C in the dark for 7 days.

To prepare the inoculum, spores were transferred from PDA slants to submerged culture medium and washed with sterile distilled water, and then diluted to approximately $3 \times 10^7$ spores/ml. The 10% (v/v) inoculum was transferred to the submerged culture medium and incubated 7 days. 10% (V/V) of the inoculum was transferred to 250 ml shark flasks containing 45 ml liquid medium and incubated for 7 days in a rotary shaker with parameters set at 30˚C and 180 rpm, respectively.

## DNA extraction

Mycelia were collected after centrifugation at 8228 ×*g* for 10 min and stored at—80˚C. Genomic DNA was extracted from mycelia using the *EasyPure*® Genomic DNA Kit (TransGen Biotech, Beijing, China) according to the manufacturer's protocol. The quantity, quality, and purity of the genomic DNA were measured using Nanodrop2000 systems and 0.8% DNase-free agarose gel electrophoresis.

## WGS and assembling

The whole genome of the *M. purpureus* CSU-M183 strain was sequenced using SMRT sequencing technology of PacBioRS II, and the sequencing quality was improved using Illumina NGS platform. The sequencing library was constructed using the TruSeq™ Nano DNA LT Sample Prep Kit–Set A (Illumina, USA) and amplified using the TruSeq PE Cluster Kit (Illumina, USA).

The quality of the assembled genome and annotated geneset were assessed first using the Benchmarking Universal Single-Copy Orthologs (version 3.1.0; BUSCO) with the fungi_odb9 dataset [33].

For raw data polymerase reads after PacBioRS II sequencing, subreads were obtained by removing the low-quality or unknown reads, adapters and duplications. The filtered reads were assembled *de novo* using the Hierarchical Genome Assembly Process (HGAP) algorithm version 2.0 [34].

For genome assembly, the default parameters of HGAP2 were used (Minimum Subread Length = 500, Minimum Polymerase Read Quality = 0.80, Minimum Polymerase Read Length = 100, Overlapper Error Rate = 0.06, Overlapper Min Length = 40) with input genome size as 30 Mb.

**PacBio library construction.** High-quality DNA (10 μg in 200 μl 10 mM Tris–HCl pH8.5) was sheared using a Covaris g-tube (Covaris Inc.) with 6000RPM for 60seconds. Sheared DNA was purified by binding to 0.45X volume of pre-washed AMPure XP beads (Beckman Coulter Inc.), and eluted in EB to >140 ng/μl. The sheared DNA was quantified on an Agilent 2100 Bioanalyzer using the 12000 kit. 5 μg of sheared DNA was end-repaired using

the PacBio DNA Template Prep Kit 2.0 (Part Number001-540-835) and incubated for 20 min at 37˚C and then 5 min at 25˚C prior to another 0.45X AMPure XP clean up, eluting in 30 μl EB. Blunt adapters were ligated before exonuclease incubation. Finally, two 0.45X AMPure bead clean ups are performed to remove enzymes and adapter dimers, and the final "SMRT bells" was eluted in 10 μl EB. Final quantification was carried out on an Agilent 2100 Bioanalyzer with 1 μl of library.

**PacBio sequencing.** The diluted library was loaded onto the instrument, along with DNA Sequencing Kit 2.0 (Part Number 100-216-400) and a SMRT Cell 8Pac. In all sequencing runs, 90 min movies were captured for each SMRT Cell loaded with a single binding complex. Primary filtering analysis was performed with the RS instrument and the secondary analysis was used the SMRT analysis pipeline version 2.1.0.

**Illumina sequencing.** Illumina library was sequenced on Hiseq X ten. Trimmomatic was used to trimm adaptor, low quality base(Q<20) and short reads(length <50bp).

## Gene prediction and annotation

AUGUSTUS [35] and SNAP [36] were performed to predict coding genes. Genome functional annotation was performed using BLASTP, as well as NCBI non-redundant (NR), SwissProt, and Protein Information Resource (PIR) protein databases. All predicted genes were classified according to the Kyoto Encyclopedia of Genes and Genomes (KEGG) metabolic pathways and Cluster of Orthologous Groups of proteins (COG).

## Prediction of secondary metabolites

To predict secondary metabolite biosynthesis of strain *M. purpureus* CSU-M183, the BGCs of secondary metabolites were annotated using antiSMASH fungi version 5.1.0 [37].

## RT-qPCR analysis

RT-qPCR was performed according to the method described by Liu et al. [21], with β-Actin as the reference gene, the genes on the MPs and citrinin gene cluster were selected, and the expression of these genes was detected by qRT-PCR during the submerged fermentation of *M. purpureus* LQ-6 and *M. purpureus* CSU-M183, respectively. For removal of residual genomic DNA, RNA samples were treated with RNase-free DNaseI (Thermo Fisher Scientific, Massachusetts, USA) following the manufacturer's protocol. The first-strand cDNA was synthesized using oligo-dT primers and EasyScript Reverse Transcriptase (TransGen Biotech, Beijing, China), according to the manufacturer's protocol. qRT-PCR was performed using the TransStartGreen qPCR SuperMix UDG (TransGen Biotech, Beijing, China) according to the manufacturer's instructions. $2^{-\Delta\Delta CT}$ was used to determine expression levels of the tested genes. The primers used in these analyses were listed in S1 Table.

## Data availability

The assembled genome sequence of *M. purpureus* CSU-M183 has been deposited into the NCBI Genbank database with an accession number of JAACNI000000000. The BioProject and BioSample information are available at PRJNA599556 and SAMN13759458, respectively. The raw sequence data of *M. purpureus* CSU-M183 has been deposited into the NCBI Genbank database with an BioProject of PRJNA824977.

### Ethical approval

This article does not contain any studies with human participants or animals performed by any of the authors.

### Statistical analysis

Each experiment was performed at least in triplicate and the results are shown as the mean ± standard deviation (SD). Statistical analyses were performed using the SPSS Statistics 23 (SPSS, Chicago, USA). Data were analyzed by one-way ANOVA, and tests of significant differences were determined by using Tukey multiple comparison or Student's t-test at $P < 0.05$.

## Results and discussion

### Overview of WGS

In the previous study, we obtained a high pigment-producing *M. purpureus* CSU-M183 strain using the ARTP mutation system [32]. The morphological characteristics of *Monascus* are closely related to the production of secondary metabolites in SF. To further study the metabolic engineering of the morphology of *M. purpureus* CSU-M183, the WGS of strain *M. purpureus* CSU-M183 was carried out. BUSCO analysis indicated 92% completeness based on fungi reference genes. Among the 290 BUSCO groups searched, 280 BUSCO groups (including complete and fragmented BUSCOs) were identified, occupying 96.5% of the total BUSCO groups. Among them, 267 groups were complete BUSCO groups, occupying 92.0% of the total BUSCO groups. The *M. purpureus* CSU-M183 genome sequence of 23.75 Mb was generated by assembling approximately 9.25 Gb raw data (353× coverage), which had a GC content of 49.13% and 69 genomic contigs (Table 1). The genome functional prediction and annotation identified 7305 protein-coding genes, with an average gene length of 1693 bp.

To investigate the functions of the coding genes and metabolic pathways, all coding sequences (CDSs) were subjected to COG and KEGG analysis [38]. The COG database (http://www.ncbi.nlm.nih.gov/COG) classifies proteins by comparing all protein sequences in the genome [39]. In total, 4157 CDSs were allocated to COG categories (Table 2), with the maximum proportion of sequences related to "carbohydrate transport and metabolism" (8.52%), followed by "amino acid transport and metabolism" (7.82%), "translation, ribosomal structure and biogenesis" (6.90%), "posttranslational modification, protein turnover, and chaperones" (6.88%), and "energy production and conversion" (5.08%). Proteins that have not been fully

**Table 1. *M. purpureus* CSU-M183 genome general features.**

| Job Metric | Value |
|---|---|
| Length of genome assembly (Mb) | 23.75 (23,752,195 bp) |
| Contig number | 69 |
| G + C content (%) | 49.4 |
| Coverage | 353X |
| Q30 (%) | 87.1 |
| SNP number | 3 |
| Number of protein-coding genes | 7305 |
| Average gene length (bp) | 1693 |
| Genes with function prediction | 5182 |
| Number of proteins with KEGG ortholog | 3362 |
| Number of proteins with COG | 4157 |

**Table 2. COG classification of predicted genes encoding proteins with annotated functions of *M. purpureus* CSU-M183 genome.**

| COG classification | Percentage(%) |
|---|---|
| RNA processing and modification | 0.77 |
| Chromatin structure and dynamics | 0.77 |
| Energy production and conversion | 5.08 |
| Cell cycle control, cell division, chromosome partitior | 2.98 |
| Amino acid transport and metabolism | 7.82 |
| Nucleotide transport and metabolism | 2.38 |
| Carbohydrate transport and metabolism | 8.52 |
| Coenzyme transport and metabolism | 4.04 |
| Lipid transport and metabolism | 4.38 |
| Translation, ribosomal structure and biogenesis | 6.90 |
| Transcription | 3.66 |
| Replication, recombination and repair | 4.40 |
| Cell wall/membrane/envelope biogenesis | 3.03 |
| Cell motility | 0.24 |
| Posttranslational modification, protein turnover, chaperones | 6.88 |
| Inorganic ion transport and metabolism | 3.80 |
| Secondary metabolites biosynthesis, transport and catabolism | 4.23 |
| General function prediction only | 19.08 |
| Function unknown | 4.81 |
| Signal transduction mechanisms | 2.36 |
| Intracellular trafficking, secretion, and vesicular transport | 2.02 |
| Defense mechanisms | 0.99 |
| Nuclear structure | 0.10 |
| Cytoskeleton | 0.77 |

identified in the genome of strain CSU-M183 were classified as "general function prediction only" (19.08%) and "function unknown" (4.81%) in COG categories.

KEGG enrichment analysis is essential for understanding the complex biological functions of genes in microorganisms, including metabolic pathways, genetic information transfer, and cytological processes [40]. Altogether, 3362 CDSs were allocated to five categories in the KEGG database, including "metabolism", "cellular process" and "environmental information processing", "genetic information processing", and "organismal systems" (Fig 1). Annotation results showed that "metabolism" is the main category of KEGG annotations (1375, 40.90%), followed by "genetic information processing" (707, 21.03%) and "organismal systems" (519, 15.44%). Moreover, CDSs were significantly enriched in "translation" (282), "carbohydrate metabolism" (279), "amino acid metabolism" (266) and "transport and catabolism" (259) sub-categories, indicating that *M. purpureus* CSU-M183 had the strong ability of protein translation, carbohydrate utilization and energy conversion.

## Identification of secondary metabolites BGCs

AntiSMASH is a widely used tool that can identify and annotate BGCs in bacterial and fungal genome sequences [41]. To further understand the biosynthesis of the secondary metabolites in strain *M. purpureus* CSU-M183, BGCs prediction of secondary metabolites were performed using antiSMASH fungi version 5.1.0. A total of 26 BGCs were detected, including terpene,

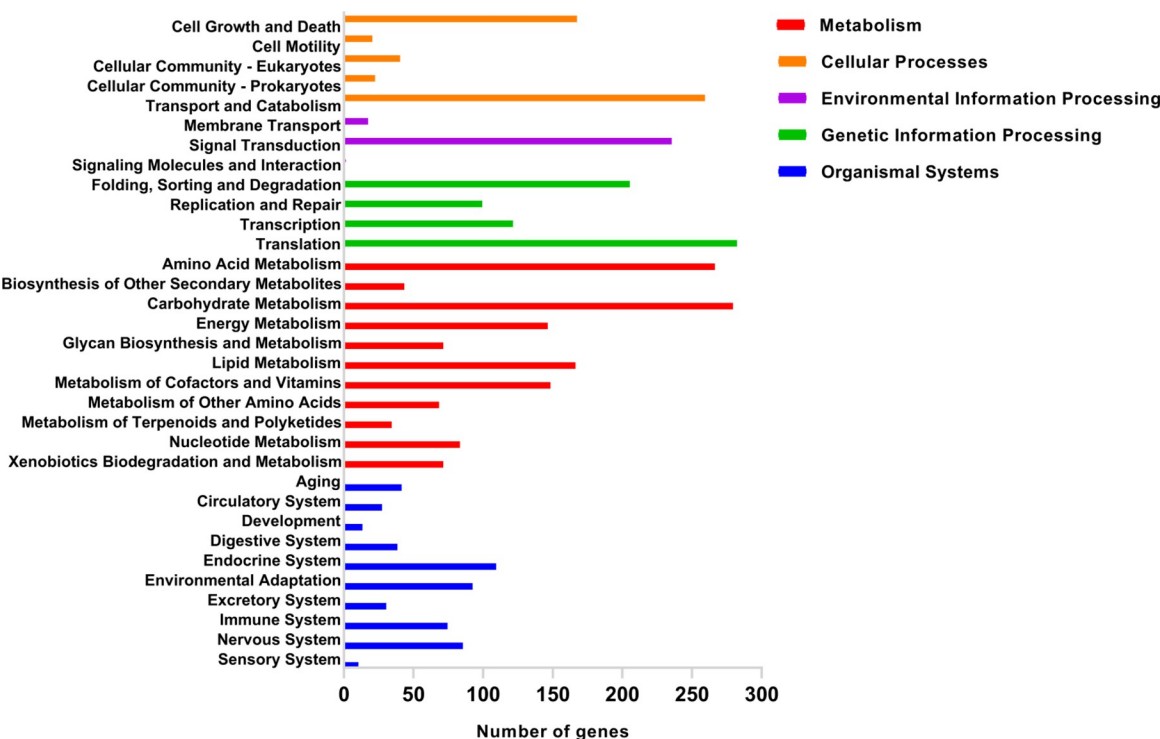

**Fig 1. Enrichment analysis of KEGG pathways for predicted genes of the *M. purpureus* CSU-M183 genome.** The y-axis represents the KEGG pathway and the x-axis denotes the number of genes.

non-ribosomal peptide synthetases (NRPS), type I polyketide synthases (T1PKS), β-lactone, and 18 putative gene clusters.

After the database search with antiSMASH, a BGC (contig 000002F, gene g3398-g3411) for citrinin within the genome sequence of strain CSU-M183 was predicted (Fig 2A), which was identical to the known citrinin BGC (GenBank accession number: AB243687.1, 21917 bp) [42], and the identity of homologous genes was 99%-100%, the predicted functions of the genes in citrinin BGC are listed in Table 3. Additionally, 81% of homologous genes were similar to those in citrinin BGC0001338, and 57% in citrinin BGC000894. A putative BGC responsible for the biosynthesis of MPs was identified in the genome of strain CSU-M183 with 41% of homologous genes showed similar to that in BGC0000027 (Fig 2B), including 16 genes (contig 000001F, gene g1401-g1416) listed in Table 4. As shown in Table 4, the identity of homologous genes was considerably high, such as gene g1409 was 96.51% similar to *Mpig*A (Polyketide synthase), gene g1407 was 95.05% similar to *Mpig*C (Ketoreductase), and gene g1406 was 95.82% similar to *Mpig*D (Acyltransferase). Moreover, by using the known monacolin K BGC (GenBank accession number: DQ176595.1, 45000 bp) of *M. pilosus* as a reference [13], no complete monacolin K BGC was detected in the genome sequence of *M. purpureus* CSU-M183 (Fig 2C). All protein-coding genes in the genome sequence were analyzed by BLASTP, where gene g3061, g2167, g4491, g1403, g1402, g4228, g1429, g1395 were homologous to the genes mkA~mkI, respectively. However, these genes do not locat in a gene cluster in genome, and the homologous protein identities were low, especially gene g1403 (with *mokD* of *M. pilosus* with 25.00% identity) (Table 5). It has been reported that overexpression of *mokD* significantly enhanced the production of monacolin K by 200.8%, which illustrated this gene play a vital role in the synthesis of monacolin K [24]. These findings were similar to that of the parent strain *M. purpureus* LQ-6, which could not produce monacolin K [43].

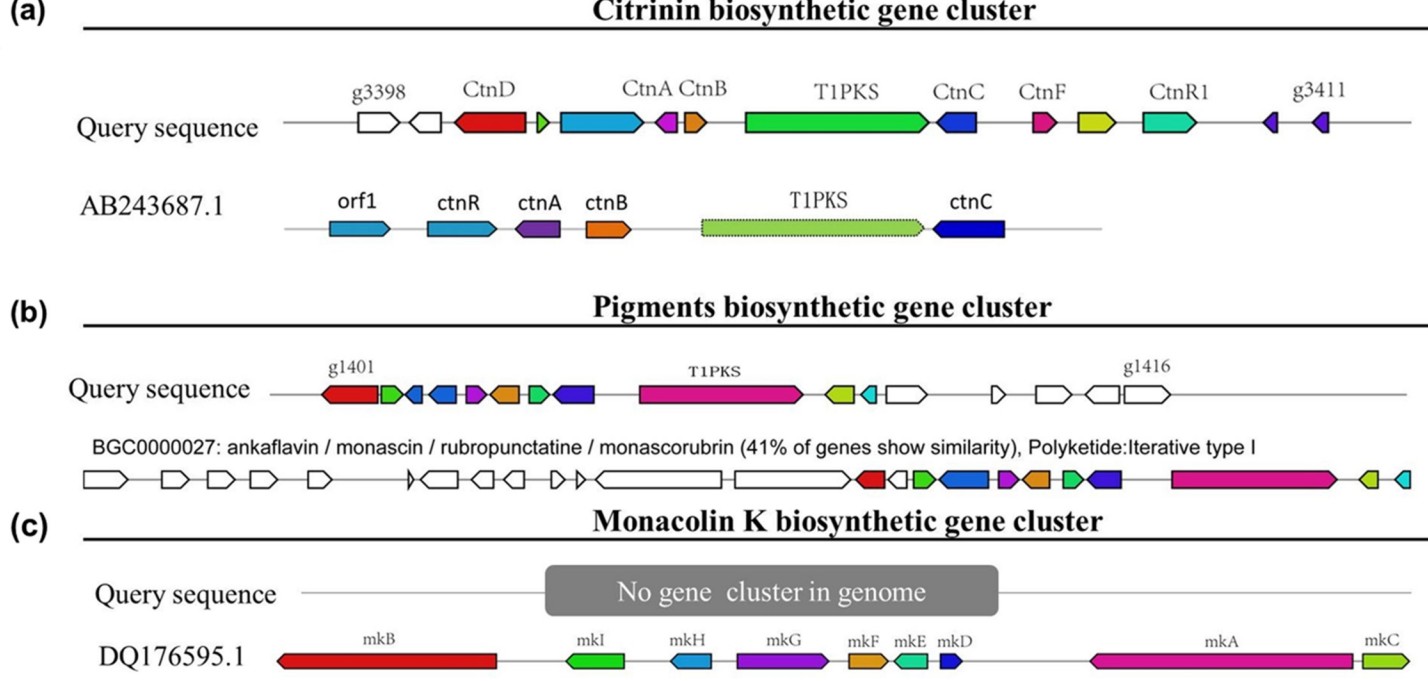

**Fig 2. Schematic representation of predominant secondary metabolites BGCs in the genome sequence of *M. purpureus* CSU-M183.** (a) Citrinin. (b) MPs. (c) Monacolin K.

Due to various species of *Monascus* and large gaps in available biological information, the development of basic theoretical research on *Monascus* has been relatively slow. With the continuous development of sequencing technology and bioinformatics, breakthrough progress has been made in biosynthetic pathways of the secondary metabolites of *Monascus* [19, 44, 45], among which MPs, citrinin and monacolin K are the most notable. However, studies on functional and comparative genomics—such as the annotation of unknown sequences, investigation of gene models, comparison of multiple sequence alignment analysis, and metabolic

**Table 3. Functional prediction of genes detected in the citrinin BGC of *M. purpureus* CSU-M183.**

| CDS | Length (bp) | Product | KO |
|---|---|---|---|
| gene = g3398 | 1762 | Unnamed protein product | |
| gene = g3399 | 1389 | Hypothetical protein, *orf7* | |
| gene = g3400 | 2796 | *ctn*D | K00108 |
| gene = g3401 | 403 | Predicted protein | K00108 |
| gene = g3402 | 3076 | Dehydrogenase, *orf1* | |
| gene = g3403 | 987 | citrinin biosynthesis oxygenase, *ctn*A | K13821 |
| gene = g3404 | 940 | citrinin biosynthesis oxydoreductase, *ctn*B | |
| gene = g3405 | 7780 | citrinin polyketide synthase, *pks*CT | |
| gene = g3406 | 1497 | citrinin biosynthesis transporter, *ctn*C | |
| gene = g3407 | 829 | *ctn*F | |
| gene = g3408 | 1498 | Hypothetical protein, *orf*8 | |
| gene = g3409 | 2116 | *ctn*R | |
| gene = g3410 | 663 | *ctn*G | |
| gene = g3411 | 567 | *ctn*G | K01673 |
| gene = g3412 | 3163 | *ctn*I | |

engineering of *Monascus* morphology, which can elaborate the relationship among secondary metabolite productivity, growth, and morphology in SF, are less.

The key gene PKS with 7838 bp responsible for citrinin biosynthesis was first identified from *M. purpureus* in 2005 [46] and then five genes encoding Zn(II)$_2$Cys$_6$ transcriptional activator, membrane transporter, dehydrogenase, oxygenase, and oxidoreductase for citrinin biosynthesis were cloned [47]. Microbial PKSs have been mainly classified into three types–type I PKSs (modular type I PKSs and iterative type I PKSs), type II PKSs, and type III PKSs. The PKS responsible for citrinin biosynthesis belongs to the iterative type I PKSs, which contains putative domains for ketosynthase (KS), acyltransferase (AT), ketoreductase (KR), dehydratase (DH), enoyl reductase (ER), methyltransferase (MT), thioesterase (TE), and acyl carrier protein (ACP) [46]. The KS domain catalyzes the condensation of precursors to extend the polyketone chain, whereas the AT domain selects the precursors, and the ACP domain makes covalent bonds between the precursors and intermediates, which are necessary for the functioning of most PKSs [48]. In 2012, the citrinin BGC with the length of 43 kb from *Monascus aurantiacus* was first published [49], including 16 open reading frames (ORFs) for *ctn*D, *ctn*E, *orf*6, *orf*1, *ctn*A, *orf*3, *orf*4, *pks*CT, *orf*5, *ctn*F, *orf*7, *ctn*R, *orf*8, *ctn*G, *ctn*H, and *ctn*I, which are dramatically similar to those of the citrinin BGC of strain *M. purpureus* CSU-M183. These results revealed high homology of citrinin BGC in *Monascus*, especially the key gene PKS. In 2012, a putative 53 kb MP BGC of *M. ruber* was first reported, which consisted of genes encoding PKSs, fatty acid synthases, regulatory factors, and dehydrogenase [3]. Xie et al. reported that gene *pigR* (gene g1401 in Table 4) is a positive regulatory gene in MPs biosynthesis pathway [50], whereas gene *MpigE* (gene g1402 in Table 4) may be involved in the conversion of different MPs [51]. The genome size of *M. purpureus* was found to be smaller than that of related filamentous fungi, indicating a significant loss of genes [19]. A previous study reported that monacolin K cannot be produced due to the lack of monacolin K biosynthesis locus in some *M. purpureus* genomes [52]. After the prediction of monacolin K BGC in the genome of strain CSU-M183, we found that there was no complete monacolin K BGC in the strain CSU-M183, which was consistent with previous studies [43, 52]. Undoubtedly, the identification of BGCs has greatly facilitated the understanding of the biosynthetic pathways of

**Table 4. Functional prediction of genes detected in the MPs BGC of *M. purpureus* CSU-M183.**

| CDS | Length (bp) | Product | Identity (%) | KO |
|---|---|---|---|---|
| gene = g1401 | 2388 | Fungal specific transcription factor, *Mpig*R | 91.41 | |
| gene = g1402 | 1108 | Alcohol dehydrogenase, *Mpig*E | 95.39 | |
| gene = g1403 | 819 | Amino oxidase/esterase, *Mpig*F | 93.01 | |
| gene = g1404 | 1395 | Amino oxidase/esterase, *Mpig*F | 94.03 | |
| gene = g1405 | 1027 | Dehydrogenase, *Mpig*H | 97.66 | |
| gene = g1406 | 1449 | Acyltransferase, *Mpig*D | 95.82 | K22889 |
| gene = g1407 | 892 | Ketoreductase, *Mpig*C | 95.05 | K11165 |
| gene = g1408 | 1719 | Citrinin biosynthesis transcriptional activator CtnR | 95.48 | |
| gene = g1409 | 8071 | Polyketide synthase, *Mpig*A | 96.51 | |
| gene = g1410 | 1032 | Peptidyl-prolyl cis-trans isomerase Cpr7 | 98.68 | K05864 |
| gene = g1411 | 573 | D-tyrosyl-tRNA(Tyr) deacylase, *Mpig*O | 95.58 | K07560 |
| gene = g1412 | 1849 | TPA: CCR4-NOT transcription complex, subunit 3 | 84.23 | K12580 |
| gene = g1413 | 634 | $C_2H_2$ finger domain protein | 76.40 | |
| gene = g1414 | 1531 | Dihydrolipoamide dehydrogenase | 84.17 | K00382 |
| gene = g1415 | 1665 | Fatty acid desaturase | 73.58 | K13076 |
| gene = g1416 | 1792 | Phosphoglucomutase | 68.45 | K01835 |

**Table 5. Functional prediction of genes detected in the monacolin K BGC of *M. purpureus* CSU-M183 by NCBI-BLASTP.**

| CDS | Length (aa) | Homologue | Homologue Length (aa) | Identity |
|---|---|---|---|---|
| gene = g3061 | 3945 | ABA02239.1_8 [gene = mkA] | 3075 | 37.19% |
| gene = g2167 | 2593 | ABA02240.1_1 [gene = mkB] | 2547 | 38.62% |
| gene = g4491 | 499 | ABA02241.1_9 [gene = mkC] | 524 | 80.53% |
| gene = g1403 | 273 | ABA02242.1_7 [gene = mkD] | 263 | 25.00% |
| gene = g1402 | 369 | ABA02243.1_6 [gene = mkE] | 360 | 39.17% |
| gene = g4228 | 1133 | ABA02245.1_4 [gene = mkG] | 1052 | 52.61% |
| gene = g1429 | 426 | ABA02246.1_3 [gene = mkH] | 455 | 78.10% |
| gene = g1395 | 570 | ABA02247.1_2 [gene = mkI] | 543 | 54.21% |

secondary metabolites in *Monascus*, which can provide theoretical support for industrial production of *Monascus* secondary metabolites.

## Expression level of Monascus pigments and citrinin clusters located genes

After 7 days of SF, the MPs and citrinin yields of *M. purpureus* LQ-6 and *M. purpureus* CSU-M183 were 43.97 U/ml, 1.27 mg/L and 83.77 U/ml, 5.34 mg/L, respectively (Fig 3A). To verify the effect of mutagenesis on the metabolism of MPs and citrinin, the relative expression levels of several key genes, *MpigA*, *MpigR*, *MpigC*, *MpigD*, *MpigE*, *MpigF*, *MpigG*, *MpigH*, *MpigI*, *MpigJ*, *MpigK*, *MpigL*, *MpigM*, *MpigP*, *MpigQ*, *cit S*, *cit A*, *cit B*, *cit C*, *cit D* and *cit E* were vestigated using RT-qPCR. As shown in Fig 3B and 3C, the relative expression levels of *MpigM*, *MpigP*, *cit B*, *cit C*, *cit D* and *cit E* in *M. purpureus* CSU-M183 were extremely significant ($p < 0.001$) compared to that of *M.purpureus* LQ-6 at 4th day; while the relative expression levels of *MpigA*, *MpigJ*, *MpigK*, *cit S*, *cit C* in *M. purpureus* CSU-M183 were extremely significant ($p < 0.001$) than that of *M. purpureus* LQ-6 at 7th day. During the fermentation process, in addition to the relative expression of extremely significant ($p < 0.001$) genes, the expression levels of other genes in the MPs and citrinin biosynthesis gene clusters in CSU-M183 were very significant ($p < 0.01$), such as *MpigA*, *MpigC*, *MpigD*, *MpigF*, *MpigG* and *cit S* at 4[th] day and *MpigD*, *MpigG*, *MpigH* and *cit B* at 7[th] day. The production of MPs and citrinin is directly or indirectly related to the function of genes in their biosynthetic gene clusters, and the relative expression of genes can directly reflect the contribution of genes in the fermentation process. A number of studies have performed functional analysis of MPs and citrinin gene clusters, such as: inactivating *MpigA* in *M. ruber*, *Monascus* lost its pigment production ability, which proved that PKS was involved in pigment synthesis [53]. MrpigJ(encoded by *MrpigJ*, a homolog of *MpigJ*) and MrpigK(encoded by *MrpigK*, a homolog of *MpigK*) form two subunits of the specialized fungal FAS, which produce the fatty acyl portion of the side chain of MPs [54]. Moreover, MrpigM, as an o-acetyltransferase, synthesized an O-11 acetyl intermediate in Chen et al's *Monascus* model, and knocking out *MrpigM*(the homolog of *MpigM*) blocked the pathway of pigment synthesis intermediate [54]. The inactivation of genes in citrinin biosynthesis gene cluster led to a significant decline on citrinin production, even lower than the detection level, such as knocking out *cit A*, *pksCT* and *cit B* [42, 55]. It indicates that the increase of MPs and citrinin production may be caused by the increase of gene expression level in gene cluster caused by ARTP mutation, and these genes are very important for the synthesis of MPs and citrinin.

## Analysis of the chitin biosynthesis pathway

As the main component of the fungal cell wall, chitin is important for the morphology of fungi. Based on the homology of amino acid sequence, chitin synthetases can be divided into

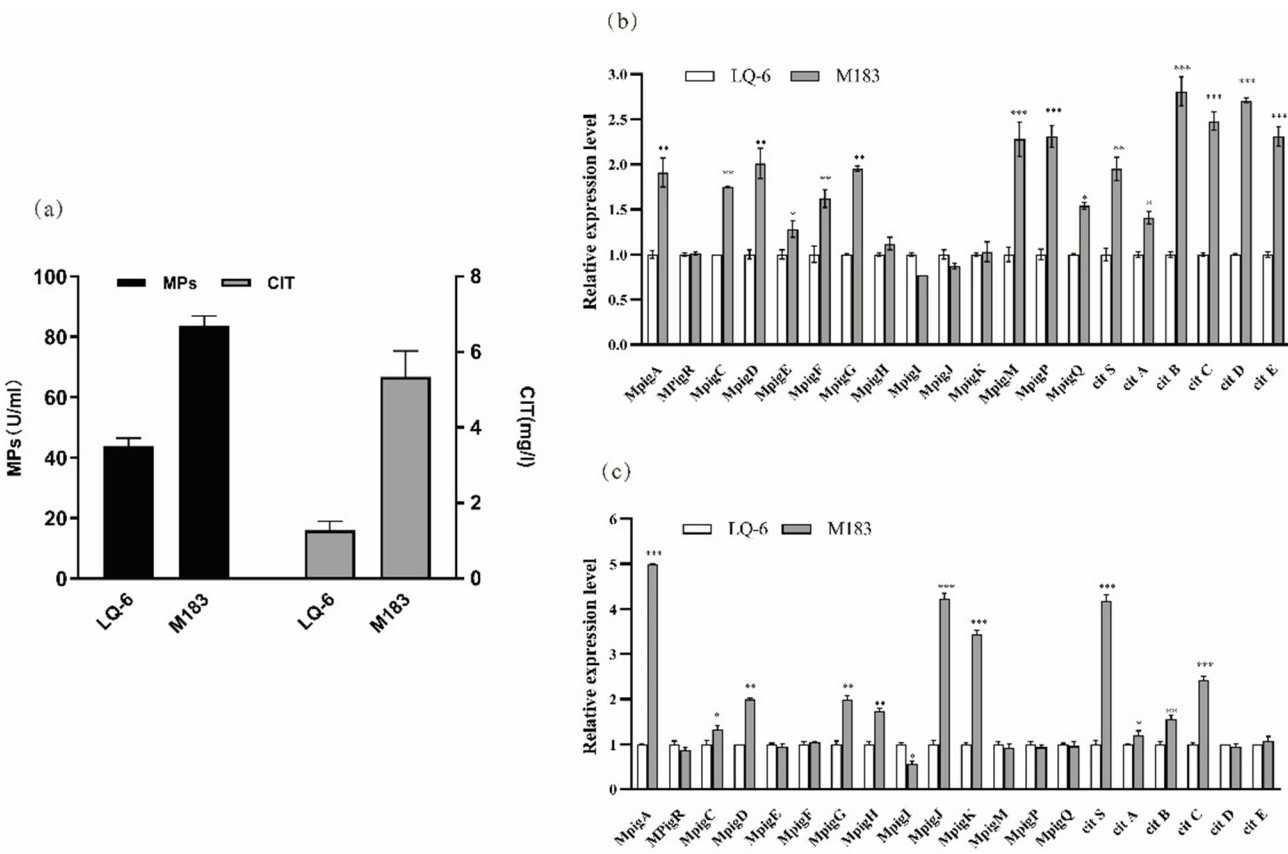

**Fig 3.** (a) Production of MPs and citrinin in SF of *M. purpureus* LQ-6 and *M. purpureus* CSU-M183 for 7 days. (b) Expression levels of genes related to MPs and citrinin biosynthesis of *M. purpureus* LQ-6 (control) and *M. purpureus* CSU-M183 at 4th day. (c) Expression levels of genes related to MPs and citrinin biosynthesis of *M. purpureus* LQ-6 (control) and *M. purpureus* CSU-M183 at 7th day. * $p < 0.05$, ** $p < 0.01$, *** $p < 0.001$.

three categories (class I-III) in *Saccharomyces cerevisiae*, four (class I-IV) in *Candida albicans*, and seven (class I-VII) in filamentous fungi. The numbers of gene encoding chitin synthetase in various filamentous fungi are different, generally containing 6–10 genes encoding chitin synthetase [56].

To date, information about chitin biosynthesis in *M. purpureus* has not been reported. To lay a foundation for the further study of morphological metabolism of *M. purpureus*, we analyzed the chitin biosynthesis of strain CSU-M183 and annotated the function of the relevant genes in the pathway. By matching the predicted chitin biosynthesis-related enzymes in CSU-M183 strain genome with the KEGG database, the biosynthetic pathway of chitin in *M. purpureus* was identified. As shown in Fig 4, phosphoacetylglucosamine mutase (PGM3) [EC:5.4.2.3] (encoded by gene g4907) converts N-acetyl-D-glucosamine 6-phosphate (GlcNAc-6P) to N-acetyl-alpha-D-glucosamine 1-phosphate (GlcNAc-1P), which is then dephosphorylated by UDP-N-acetylglucosamine diphosphorylase (UAP1) [EC:2.7.7.23] (encoded by gene g6630) to yield UDP-N-acetyl-D-glucosamine (UDP-GlcNAc). Moreover, chitin synthase (*chs*1) [EC:2.4.1.16] (encoded by genes: g872, g920, g3078, and g5640) converts UDP-GlcNAc to chitin. Additionally, the other genes encoding the important enzymes in the biosynthetic pathway of chitin were annotated, such as N-acetylmuramic acid 6-phosphate etherase (murQ) [EC:4.2.1.126] encoded by gene g1905, chitinase [EC:3.2.1.14] encoded by genes g3222, g6372 and g1142, and glucosamine-phosphate N-acetyltransferase (GNPNAT1, GNA1) [EC:2.3.1.4] encoded by gene g2832.

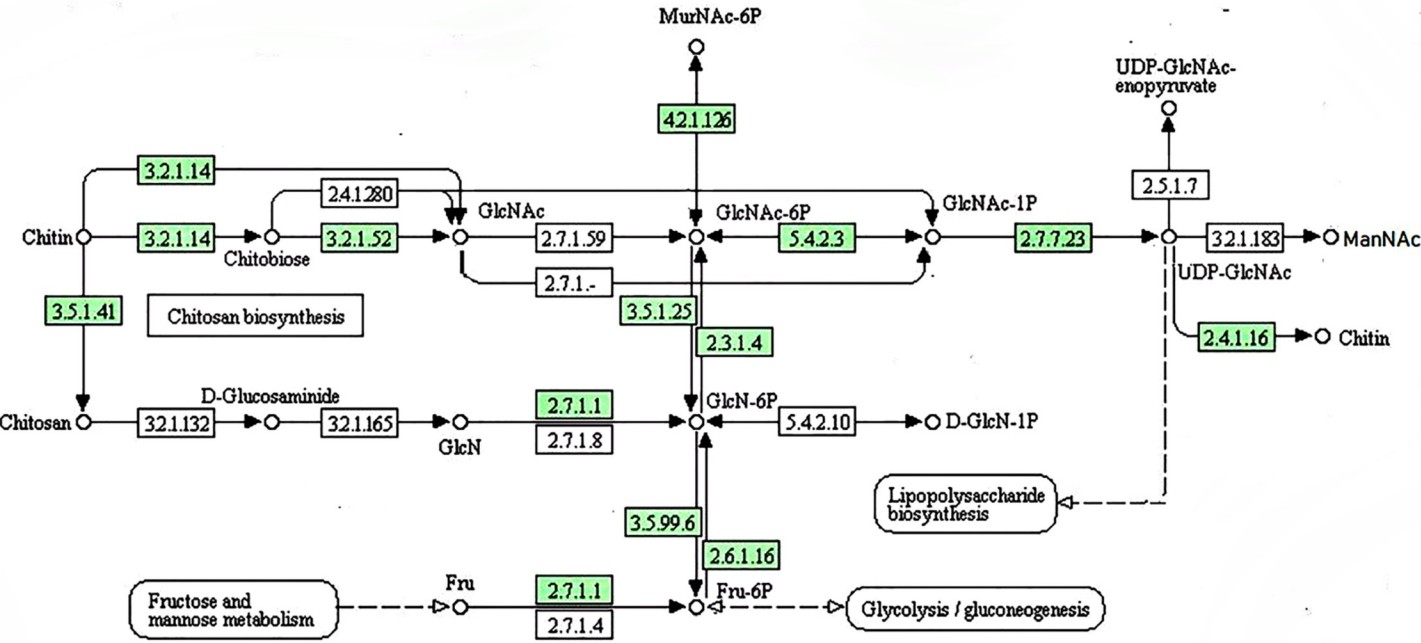

**Fig 4. Prediction of chitin biosynthetic pathway in *M. purpureus* CSU-M183 genome.** The corresponding enzymes involved in each bioconversion step are shown in green.

Class III chitin synthases only exist in the cell wall of fungi with high chitin content and are essential for regulating the mycelial aggregation morphology of fungi. Gene *chs*B in *Aspergillus fumigatus* plays an important role in cell wall biosynthesis, hyphal growth, and asexual reproduction [57]. To provide genetic resources for further studies, we mainly identified the gene *chs*B (gene g4739, 3243 bp) encoding class III chitin synthase in *M. purpureus* CSU-M183.

Generally, the specific characteristic of their SF is the aggregation of mycelia that are affected by environmental conditions, leading to different rheological properties of the fermentation broth. Such changes affect the transfer of mass, heat, and momentum, as well as the biosynthesis and production efficiency of target products. Moreover, the morphology of hyphae is closely related to the biosynthesis of secondary metabolites, and changes in the mycelium morphology of *Monascus* can regulate the level of secondary metabolites [17, 24]. As the main component of the fungal cell wall, chitin affects the mycelial morphological changes such as apical extension, branch growth, and differentiation. Blocking the biosynthetic pathway of chitin inevitably changes the mycelial aggregation and regulates metabolic pathways of target products. With the rise of the SF technology of *Monascus*, the effects of mycelial morphology on the biosynthesis of secondary metabolites in the fermentation process have attracted much attention. However, research on the metabolic engineering of *Monascus* morphology is still in the blank stage. In this article, we commented the strategies for morphological regulation of filamentous fungi, and discussed the impact of calcium signal transduction and chitin biosynthesis on apical hyphal growth and mycelial branching. Furthermore, based on the WGS analysis of strain *M. purpureus* CSU-M183, we will use genetic engineering technology to disturb the chitin biosynthesis of *M. purpureus* CSU-M183, change the mycelial aggregation morphology in the process of SF, regulate the biosynthesis of secondary metabolites, and clarify the molecular mechanism of the regulation of morphological on secondary metabolism using genetic engineering technology and histochemical correlation analysis.

## Conclusion

Genomic information of *M. purpureus* CSU-M183 reported here can serve as a reference genome for *Monascus* genomics research. It's predicted that the secondary metabolites BGCs and the chitin biosynthetic pathway in the genome of *M. purpureus* CSU-M183. We verified that ARTP induced significantly the upregulated expression of most *Monascus* pigment and citrinin clusters located genes by RT-qPCR. In addition, we annotated and classified the chitin biosynthesis genes of *M. purpureus* CSU-M183, which offer a strategy of morphological metabolic engineering. In conclusion, we provided genomic resources for further biological studies on the metabolic engineering of the morphology of *Monascus*.

## Supporting information

**S1 Table. The primers used in this study.**
(DOCX)

## Acknowledgments

We would like to thank Editage for English language editing.

## Author Contributions

**Data curation:** Jun Liu.

**Formal analysis:** Song Zhang.

**Methodology:** Xiaofang Zeng.

**Resources:** Qinlu Lin.

**Writing – original draft:** Song Zhang.

**Writing – review & editing:** Jun Liu.

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
