## [Decision Letter · Decision Letter 0]

22 Feb 2022

PONE-D-22-01622Analysis of secondary metabolite gene clusters and chitin biosynthesis pathways of Monascus purpureus with high production of pigment and citrinin based on whole-genome sequencingPLOS ONE

Dear Dr. Liu,

Thank you for submitting your manuscript to PLOS ONE. After careful consideration, we feel that it has merit but does not fully meet PLOS ONE’s publication criteria as it currently stands. Therefore, we invite you to submit a revised version of the manuscript that addresses the points raised during the review process.

Two experts in the field have reviewed your manuscript, you can find their comments below and in a separate pdf file. They make valuable suggestions to improve the manuscript. Please revise your manuscript carefully according to the reviewers' suggestions, except for the question of annotation where reviewer 1 suggests using MAKER instead of an Augustus/SNAP-based pipepline. Changing the annotation pipeline might be something to keep in mind for future studies. Please also make sure that all your data including the raw sequence reads have been submitted to the appropriate databases. The BioProject entry for your project suggests that you have submitte the assembly, but perhaps not yet the raw sequence reads (they could be submitted, for example, to the NCBI SRA database).

We look forward to receiving your revised manuscript.

Kind regards,

Minou Nowrousian

Academic Editor

PLOS ONE

Journal Requirements:

e would also like to thank Editage for English language editing. This work was supported by the National Natural Science Foundation of China (No. 32101906), Natural Science Foundation of Hunan Province (No. 2021JJ31146); Open Project Program of the Hunan Provincial Key Laboratory of Food Safety Monitoring and Early Waring (No. 2021KFJJ02), Education Department of Scientific Research Project of Hunan Province (No. 20B619) and Education Department of Postgraduate Research and Innovation Project of Hunan Province（No. CX20210863, CX20210898）

This work was supported by the National Natural Science Foundation of China (No. 32101906), Natural Science Foundation of Hunan Province (No. 2021JJ31146); Open Project Program of the Hunan Provincial Key Laboratory of Food Safety Monitoring and Early Waring (No. 2021KFJJ02), Education Department of Scientific Research Project of Hunan Province (No. 20B619) and Education Department of Postgraduate Research and Innovation Project of Hunan Province（No. CX20210863, CX20210898）

e would also like to thank Editage for English language editing. This work was supported by the National Natural Science Foundation of China (No. 32101906), Natural Science Foundation of Hunan Province (No. 2021JJ31146); Open Project Program of the Hunan Provincial Key Laboratory of Food Safety Monitoring and Early Waring (No. 2021KFJJ02), Education Department of Scientific Research Project of Hunan Province (No. 20B619) and Education Department of Postgraduate Research and Innovation Project of Hunan Province（No. CX20210863, CX20210898）

Reviewers' comments:

Reviewer's Responses to Questions

**Comments to the Author**

1. Is the manuscript technically sound, and do the data support the conclusions?

Reviewer #1: Yes

Reviewer #2: Partly

2. Has the statistical analysis been performed appropriately and rigorously? 

Reviewer #1: Yes

Reviewer #2: I Don't Know

3. Have the authors made all data underlying the findings in their manuscript fully available?

Reviewer #1: Yes

Reviewer #2: Yes

4. Is the manuscript presented in an intelligible fashion and written in standard English?

Reviewer #1: Yes

Reviewer #2: Yes

5. Review Comments to the Author

Reviewer #1: The Monascus purpureus csu-m183 strain is of industrial importance and warrants a detailed genomic analysis. The authors here have done a good job describing its genomic contents, particularly in relation to pigment and citrinin metabolism. I have a couple of comments.

1. Annotation pipelines such as MAKER pipeline take into account various evidences to predict the genes in a genome. It might perform better than solely using augustus and SNAP.

2. In methods section, the genome assembly process is not described in enough details that it can be reproduced. Parameters used in HGAP program are missing.

3. In introduction, it is not clearly rationalized why the genome sequencing of this particular strain is necessary while several other high pigment producing strains are already sequenced.

4. With several M. purpureus genomes available, it might be helpful to conduct a comparative genomic analysis among them to elucidate likely mechanism that CSU-m183 strain is special in its metabolic capacity.

Reviewer #2: 1. 1. Is the manuscript technically sound, and do the data support the conclusions?

Ans: Concerns described in the attached document.

2. Has the statistical analysis been performed appropriately and rigorously?

Ans: I didnot find a section in the manuscript about the statistical analysis they used, specifically for the qPCR data.

6. PLOS authors have the option to publish the peer review history of their article (what does this mean?). If published, this will include your full peer review and any attached files.

Reviewer #1: No

Reviewer #2: No

---

## [Author Response · Author response to Decision Letter 0]

31 Mar 2022

ID: PONE-D-22-01622 

Title: Analysis of secondary metabolite gene clusters and chitin biosynthesis pathways of Monascus purpureus with high production of pigment and citrinin based on whole-genome sequencing

Journal: Plos One

Dear Editor,

Thank you very much for your and reviewers' comments concerning our manuscript entitled “Analysis of secondary metabolite gene clusters and chitin biosynthesis pathways of Monascus purpureus with high production of pigment and citrinin based on whole-genome sequencing（PONE-D-22-01622）”. We have studied the editor/reviewer comments and made corrections that we hope meet with your approval. The main corrections have been marked in red in the revised manuscript (PONE-D-22-01622) and our details responses to the comments are given below in bold type.

Editor' comments:

Response：Thank you very much for your comment, we have changed the manuscript to comply with PLOS ONE's style requirements, including file naming requirements

we would also like to thank Editage for English language editing. This work was supported by the National Natural Science Foundation of China (No. 32101906), Natural Science Foundation of Hunan Province (No. 2021JJ31146); Open Project Program of the Hunan Provincial Key Laboratory of Food Safety Monitoring and Early Waring (No. 2021KFJJ02), Education Department of Scientific Research Project of Hunan Province (No. 20B619) and Education Department of Postgraduate Research and Innovation Project of Hunan Province（No. CX20210863, CX20210898）

Response：Thank you for your comment. Funders JL and SZ designed the study and annotated the genome, respectively. SZ and JL co-write the manuscript. All authors read and approved the final version of the manuscript.

we would also like to thank Editage for English language editing. This work was supported by the National Natural Science Foundation of China (No. 32101906), Natural Science Foundation of Hunan Province (No. 2021JJ31146); Open Project Program of the Hunan Provincial Key Laboratory of Food Safety Monitoring and Early Waring (No. 2021KFJJ02), Education Department of Scientific Research Project of Hunan Province (No. 20B619) and Education Department of Postgraduate Research and Innovation Project of Hunan Province（No. CX20210863, CX20210898）

Response：Thank you for your comment, we have deleted the funding information in the manuscript and submitted them in the Funding Statement. 

Response：Thank you for your comment, we have provided accession numbers for the relevant data in the Materials and Methods of the manuscript (page 9, line 9-13).

Response：Thank you for your comment, we have moved the ethics statement to the Materials and Methods section of the manuscript (page 9, line 14-16).

Response：Thank you for your comment, we have changed the supplemental information as requested (page 31, line 20-21). 

Response：Thank you for your comment, we have checked the reference list to make sure it is complete and correct.

Reviewers' comments:

Reviewer #1: 

The Monascus purpureus CSU-M183 strain is of industrial importance and warrants a detailed genomic analysis. The authors here have done a good job describing its genomic contents, particularly in relation to pigment and citrinin metabolism. I have a couple of comments.

1. Annotation pipelines such as MAKER pipeline take into account various evidences to predict the genes in a genome. It might perform better than solely using augustus and SNAP.

Response：Thank you very much for your comment.

2. In methods section, the genome assembly process is not described in enough details that it can be reproduced. Parameters used in HGAP program are missing.

Response：Thank you very much for your comment, we have re-described the genome assembly process in detail in the Materials and methods section of WGS and assembling (page 7, line 4-7). 

3. In introduction, it is not clearly rationalized why the genome sequencing of this particular strain is necessary while several other high pigment producing strains are already sequenced.

Response：Thank you very much for your comment. There are not many published whole genome sequences of Monascus so far, and there are certain differences between the Monascus species, such as some do not contain a complete monacolin K biosynthetic gene cluster, and some do not produce citrinin, etc. Besides, we illustrate that the cell wall is an ideal target for the regulation of fungal morphology, chitin is the main component of cell wall, but the number of genes encoding chitin synthase and chitin biosynthesis pathway in M. purpureus is still unclear (page 5, line 3-11). Therefore, we comprehensively predicted the gene encoding chitin synthase and the biosynthetic pathway of chitin in M. purpureus CSU-M183 through whole genome sequencing for future morphological studies. 

4.With several M. purpureus genomes available, it might be helpful to conduct a comparative genomic analysis among them to elucidate likely mechanism that CSU-M183 strain is special in its metabolic capacity.

Response：Thank you very much for your suggestion. Based on the whole genome sequence of strain CSU-M183, we have performed genome resequencing of the mutant strains (morphologically different strain), focusing on the genes related to morphology, metabolism and growth, and to explore the relationship between the morphology, growth and metabolism of Monascus, for our future study.

Reviewer #2:

In this study, the authors employed a hybrid sequencing strategy (PacBio and Illumina) to get the whole-genome assembly of a Monascus purpureus mutant. The authors comprehensively addressed the predicted secondary metabolite biosynthetic clusters as well as the expression of chitin and Monascus pigments genes in mutant strain, Monascus purpureus CSU-M183. Although there are some concerns (given below), I find this study relevant for the research community.

Comments:

1. In the Materials and methods section of WGS and assembling, the information about the method of PaBioRS II library preparation is not described. The authors should add information about the PacBio sequencing library construction and parameters used for filtering the reads. The authors should also provide information on the Illumina NGS sequencing platform used, and the processing of Illumina reads.

Response：Thank you very much for you comment, we have added relavant information in the Materials and methods section of WGS and assembling (page 7 line 8 to page 8 line 6). 

2. They used two sequencing platforms to generate the assembly of Monascus purpureus mutant. However, they did not describe how they validated the completeness of the genome assembly.

Response：Thank you very much for you comment. Indeed the genome assembly was performed using HGAP2 for PacBio RSII reads, and Illumina reads were mapped on assembled genome to correct possible SNPs and Indels. Though we finally obtained a high-quality genome with 69 contigs, it’s not assembled into chromosome level. So we did not estimate the completeness of the genome (page 7 line 8 to page 8 line 6). 

3. I am confused about the description of how they obtained the M. purpureus mutant strain. If they created the mutant strain from parent M. purpureus LQ-6, the authors have to describe the mutagenesis and screening. Otherwise, they have to cite the previous study that reported the M. purpureus CSU-M183 mutant.

Response：Thank you very much for you comment, we have cited our previous research of the M. purpureus CSU-M183 mutant（page 6, line 2）.

4. Fig. 1 – I recommend giving the alphabets corresponding to each annotated function inside the pie chart. I find the pie chart shown in the manuscript not helpful because some colors are same for annotated functions. For example, the color for E, O, and Z are the same. The other option is providing the percentage of predicted genes with annotated function as a table instead of a pie chart.

Response：Thank you very much for you comment, the percentage of predicted genes with annotation functions have been presented in a table (Table 2).

5. In the methods section of RT-qPCR, the authors should add the statistical methods used for qPCR analysis. They did not mention how many replicates they used for qPCR experiments. Response：Thank you very much for you comment. We have supplemented Statistical analysis in the Materials and Methods to illustrate the statistical methods used in this study and the number of experimental replicates (page 9, line 17-22). 

6. In the RT-qPCR results, the authors should give the p-values of significantly higher qPCR expression results instead of using terminologies such as “obviously higher”, “relatively significant”, “very significant,” or “extremely significant”. I find it hard to follow what the authors meant by “extremely significant” and “very significant”. They should also describe in the legend of Figure 4 on what each type of asterisks (**, ***) meant.

Response：Thank you very much for your comment, we gave the p-values of significantly higher qPCR expression results and supplemented in the legend of Fig 3 on what each type of asterisks meant (page 22, line 12-13). 

7. Page #14 (Line #18) - Page # 15 (Line #3) – “As shown in Fig. 4, phosphoacetylglucosamine mutase (PGM3) [EC:5.4.2.3] (encoded by gene g4907) converts N-acetyl-D-glucosamine 6-phosphate (GlcNAc-6P) to N-acetyl-alpha-D glucosamine 1-phosphate (GlcNAc-1P), which is then dephosphorylated by UDP-N22 acetylglucosamine diphosphorylase (UAP1) [EC:2.7.7.23] (encoded by gene g6630) to yield UDP-N-acetyl-D-glucosamine (UDP-GlcNAc)”. The above description matches Fig 5, not Fig 4.

Response：Thank you very much for your comment，we have corrected it. 

8. This manuscript requires language editing before accepting for publication. I have noticed the repetition of words, spelling, and grammatical errors. Although some paragraphs are fairly well written, I recommend revising the manuscript with the assistance of a native English speaker. Response：Thank you for your comment, the manuscript has been revised with the help of native English speakers.

9. Authors should correct the format of in-text citation throughout the manuscript. Also, keep consistency in the units' format (Ex: ml or mL)

Response：Thank you very much for your comment, we have changed the in-text citation format as requested and unified the unit of volume to ml.

---

## [Editor Report · Decision Letter 1]

8 Apr 2022

PONE-D-22-01622R1Analysis of secondary metabolite gene clusters and chitin biosynthesis pathways of Monascus purpureus with high production of pigment and citrinin based on whole-genome sequencingPLOS ONE

Dear Dr. Liu,

Thank you for submitting your manuscript to PLOS ONE. After careful consideration, we feel that it has merit but does not fully meet PLOS ONE’s publication criteria as it currently stands. Therefore, we invite you to submit a revised version of the manuscript that addresses the points raised during the review process.

The revised manuscript is improved with respect to many aspects that were pointed out by the reviewers, but some aspects still need improvement. Please carefully address the following points: 1. Please submit the PacBio sequencing reads (raw data) and Illumina data you generated to an appropriate database (e.g. NCBI SRA) and give the SRA accession number. Currently, it appears as if you only submitted the assembled genome, but not the raw sequence data, which are also important. 2. Some sections of the manuscript were apparently changed in an attempt to improve the English, but this is not actually better in many cases. Especially the abstract now contains a number of language problems in the changes sentences that were not present before. Please carefully check the manuscript and correct language problems. 3. Reviewer 2 suggested checking for genome completeness, which is something that can and should be done especially with genomes that are not assembled fully at chromosome level. You can use the presence of conserved genes to check for completeness (of the gene space at least), e.g. by using the BUSCO tool or other methods to check for the presence of expected conserved eukaryotic genes. Please do this and document the method used and the results in your manuscript. 4. Reviewer 1 asked about the reasons why this particular Monascus purpureus strain was sequenced. This is still not sufficiently explained in the introduction. Please explain what makes this strain interesting (not the species as such, but this particular strain).

We look forward to receiving your revised manuscript.

Kind regards,

Minou Nowrousian

Academic Editor

PLOS ONE
---

## [Author Response · Author response to Decision Letter 1]

22 Apr 2022

Editor' comments:

1. Please submit the PacBio sequencing reads (raw data) and Illumina data you generated to an appropriate database (e.g. NCBI SRA) and give the SRA accession number. Currently, it appears as if you only submitted the assembled genome, but not the raw sequence data, which are also important. 

Response：Thank you very much for your comment. Firstly, due to the COVID-19 in Shanghai, the people were strongly asked to work at home, and the raw date is being stored on the server of sequencing company (Chinese National Human Genome Center at Shanghai). Secondly, we would submit the raw date to NCBI-SRA as soon as possible when the staff return to work. Thus, we just provided the raw date of SRA acceptance number with BioProject of PRJNA824977 in this revised manuscript (PONE-D-22-01622R2；Page 9，Line 14-15).

2. Some sections of the manuscript were apparently changed in an attempt to improve the English, but this is not actually better in many cases. Especially the abstract now contains a number of language problems in the changes sentences that were not present before. Please carefully check the manuscript and correct language problems.

Response：Thank you for your comment, the manuscript has been revised and marked in blue.

Reviewers' comments:

Reviewer #1: 

1. Reviewer 1 asked about the reasons why this particular Monascus purpureus strain was sequenced. This is still not sufficiently explained in the introduction. Please explain what makes this strain interesting (not the species as such, but this particular strain).

Response：Thank you very much for your comment. We have explained what makes this strain interesting in the introduction（Page 5, Line 6-16）. 

Reviewer #2:

Reviewer 2 suggested checking for genome completeness, which is something that can and should be done especially with genomes that are not assembled fully at chromosome level. You can use the presence of conserved genes to check for completeness (of the gene space at least), e.g. by using the BUSCO tool or other methods to check for the presence of expected conserved eukaryotic genes. Please do this and document the method used and the results in your manuscript.

Comments:

Response：Thank you very much for you comment. We have checked the completeness of the genome by using the BUSCO tool. The corresponding methods（Page 6, Line 20-22） and results（Page 10, Line 9-13）have been supplemented in the Manuscript.

---

## [Editor Report · Decision Letter 2]

25 Apr 2022

Analysis of secondary metabolite gene clusters and chitin biosynthesis pathways of Monascus purpureus with high production of pigment and citrinin based on whole-genome sequencing

PONE-D-22-01622R2

Dear Dr. Liu,

We’re pleased to inform you that your manuscript has been judged scientifically suitable for publication and will be formally accepted for publication once it meets all outstanding technical requirements.

Kind regards,

Minou Nowrousian

Academic Editor

PLOS ONE
---

## [Editor Report · Acceptance letter]

19 May 2022

PONE-D-22-01622R2 

Analysis of secondary metabolite gene clusters and chitin biosynthesis pathways of *Monascus purpureus* with high production of pigment and citrinin based on whole-genome sequencing 

Dear Dr. Liu:

I'm pleased to inform you that your manuscript has been deemed suitable for publication in PLOS ONE. Congratulations! Your manuscript is now with our production department. 

Kind regards, 

on behalf of

Dr. Minou Nowrousian 

Academic Editor

PLOS ONE